# An Increase in HSF1 Expression Directs Human Mammary Epithelial Cells toward a Mesenchymal Phenotype

**DOI:** 10.3390/cancers15204965

**Published:** 2023-10-12

**Authors:** Natalia Vydra, Agnieszka Toma-Jonik, Patryk Janus, Katarzyna Mrowiec, Tomasz Stokowy, Magdalena Głowala-Kosińska, Damian Robert Sojka, Magdalena Olbryt, Wiesława Widłak

**Affiliations:** 1Maria Skłodowska-Curie National Research Institute of Oncology, Gliwice Branch, Wybrzeże Armii Krajowej 15, 44-102 Gliwice, Poland; agnieszka.toma-jonik@gliwice.nio.gov.pl (A.T.-J.); patryk.janus@gliwice.nio.gov.pl (P.J.); katarzyna.mrowiec@gliwice.nio.gov.pl (K.M.); magdalena.glowala-kosinska@gliwice.nio.gov.pl (M.G.-K.); damian.sojka@gliwice.nio.gov.pl (D.R.S.); magdalena.olbryt@gliwice.nio.gov.pl (M.O.); 2Scientific Computing Group, IT Division, University of Bergen, N-5008 Bergen, Norway; tomasz.stokowy@k2.uib.no

**Keywords:** epithelial-to-mesenchymal transition, heat shock transcription factor 1, HSF1 inhibitor, human mammary epithelial cells

## Abstract

**Simple Summary:**

Epithelial, compactly packed cells form a protective layer lining the surfaces of organs and cavities throughout the body. They can lose their features and gain migratory and invasive (mesenchymal) properties. This epithelial-to-mesenchymal transition (EMT) is essential for numerous developmental processes and wound healing but can also cause organ fibrosis and metastasis initiation in cancer progression. We found that the heat shock transcription factor 1 (HSF1), commonly activated by environmental stress, may contribute to EMT in the human mammary epithelial cells. Moreover, elevated HSF1 levels correlate with invasive features of breast cancer cells, and the use of the HSF1 inhibitor DTHIB significantly inhibits their growth. We postulate that HSF1 may be involved in the remodeling of the mammary gland architecture over the female lifetime, as well as the acquisition of invasive cancer cell phenotype. Therefore, HSF1 inhibition could be tested as an adjuvant treatment for breast cancer patients.

**Abstract:**

HSF1 is a well-known heat shock protein expression regulator in response to stress. It also regulates processes important for growth, development or tumorigenesis. We studied the HSF1 influence on the phenotype of non-tumorigenic human mammary epithelial (MCF10A and MCF12A) and several triple-negative breast cancer cell lines. MCF10A and MCF12A differ in terms of HSF1 levels, morphology, growth in Matrigel, expression of epithelial (CDH1) and mesenchymal (VIM) markers (MCF10A are epithelial cells; MCF12A resemble mesenchymal cells). HSF1 down-regulation led to a reduced proliferation rate and spheroid formation in Matrigel by MCF10A cells. However, it did not affect MCF12A proliferation but led to CDH1 up-regulation and the formation of better organized spheroids. HSF1 overexpression in MCF10A resulted in reduced CDH1 and increased VIM expression and the acquisition of elongated fibroblast-like morphology. The above-mentioned results suggest that elevated levels of HSF1 may direct mammary epithelial cells toward a mesenchymal phenotype, while a lowering of HSF1 could reverse the mesenchymal phenotype to an epithelial one. Therefore, HSF1 may be involved in the remodeling of mammary gland architecture over the female lifetime. Moreover, HSF1 levels positively correlated with the invasive phenotype of triple-negative breast cancer cells, and their growth was inhibited by the HSF1 inhibitor DTHIB.

## 1. Introduction

The epithelium, one of the four main tissue types in the human body, is formed by cells, which are closely bound to each other to ensure the mechanistic integrity of the tissues and the permeability barrier. It usually lines the outer surfaces of the organs and blood vessels, the inner surfaces of cavities, and it also covers the outer surface of the human body (skin). Nevertheless, during development (e.g., embryogenesis), epithelial cells may give rise to mesodermal tissue (among others, mesenchyme), which involves the profound modification of epithelial cell phenotypes. These modifications are exhibited by changes in cell shape and polarity, delamination and migration, which are characteristic of mesenchymal cells. This transition between epithelial and mesenchymal states is known as the epithelial-to-mesenchymal transition (EMT) and plays an essential role in morphogenesis during embryogenesis, wound healing, cancer progression and metastasis formation [1].

The EMT plays an important role in breast development. The mammary gland (breast) undergoes dynamic changes in tissue architecture and function over the female lifetime. It is composed of branching epithelial ducts embedded within a stromal fat, comprising adipocytes, fibroblasts, lymphatic cells and vascular cells, whose interactions ensure proper organ functioning. The mammary ductal epithelium is a bi-layered structure, which includes two types of cells: luminal and basal/myoepithelial cells. The luminal cells are polarized, cuboid in shape and surround a lumen. They can differentiate into either ductal cells or milk-producing cells. The basal/myoepithelial cells are spindle-shaped and located in a basal position adjacent to the basement membrane. Furthermore, there is evidence of a mammary epithelial cell hierarchy, where bi-potent or multi-potent stem cells give rise to more differentiated cell progenitors, essential for breast growth and development throughout a woman’s life [2,3]. The major developmental changes within the mammary gland occur postnatally and begin in puberty. At this stage, the branched epithelial network of the ducts is formed, and the appearance of motile cells harboring EMT features is correlated with tube elongation [4]. EMT is also observed in many models of breast cancer. This so-called oncogenic EMT is associated with pro-metastatic characteristics, such as increased motility, invasiveness, anoikis resistance and evasion of the immune system. Additionally, the acquisition of cancer stem cell-like properties can be linked with EMT and is associated with therapeutic resistance [5,6].

Down-regulation of E-cadherin (CDH1) is a hallmark of EMT. This is associated with the destabilization of adherens junctions and diffused cytoplasmic and nuclear localization of β-catenin, which promotes the expression of mesenchymal proteins, such as N-cadherin (CDH2), vimentin (VIM) and fibronectin (FN1). This process is orchestrated by complex regulatory networks involving transcriptional control with SNAI1 and SNAI2, ZEB1 and ZEB2, TWIST1 among transcriptional factors, non-coding RNAs, chromatin remodeling and epigenetic modification [7].

Some data indicate that the heat shock factor 1 (HSF1), a well-known regulator of stress response, can promote TGFβ-induced EMT in the ovarian cancer model [8] or in the breast cancer model [9,10]. HSF1 is a ubiquitously expressed transcription factor. It mainly regulates the expression of stress-inducible heat shock proteins (HSPs), which function as molecular chaperones assisting protein folding. In mammals, HSF1 deficiency leads to the dysregulation of many physiological functions (e.g., thermotolerance, systemic body temperature regulation, redox homeostasis, antioxidative defenses, immune response, motor activity, smell, hearing, memory and others), also related to gametogenesis and reproduction [11,12,13]. HSF1 also plays an important role in cancerogenesis. It can support tumor initiation and growth, as well as metastasis formation and angiogenesis [14,15]. Although many functions of HSF1 in the regulation of acute stress response are established, several questions regarding HSF1 activity during development remain unresolved.

Our objective was to investigate the function of HSF1 in modulating the plasticity of human mammary epithelial cells, and we demonstrated that HSF1 can affect the intrinsic ability of these cells to acquire mesenchymal characteristics. Furthermore, we analyzed the effect of HSF1 on the phenotype of triple-negative breast cancer cells and found that increasing HSF1 levels positively correlated with the acquisition of an invasive phenotype in Matrigel.

## 2. Materials and Methods

### 2.1. Cell Culture and Treatment

Non-tumorigenic human breast epithelial cell lines, MCF10A (authenticated in 2016) and MCF12A (ATCC, Manassas, VA, USA), were cultured in DMEM/F12 medium supplemented with 5% horse serum (BioWest, Nuaillé, France), 5 µg/mL insulin (Sigma-Aldrich, Saint Louis, MO, USA), 0.5 µg/mL hydrocortisone (Sigma-Aldrich), 20 ng/mL EGF (Sigma-Aldrich). Breast cancer MDA-MB-468, BT-549 and HCC1395 cells were cultured in RPMI-1640 medium supplemented with 10% Fetal Bovine Serum (EURx, Gdansk, Poland). MDA-MB-231 and CAL120 breast cancer cells were cultured in high glucose DMEM medium supplemented with 10% FBS (EURx). Cells were routinely tested for mycoplasma contamination. Heat shock was carried out by placing plates with logarithmically growing cells in a water bath at a temperature of 43 °C for 1 h. Subsequently, cells were allowed to recover for the indicated time in a CO_2_ incubator at 37 °C. DTHIB was added at the indicated concentration 48 h before heat shock.

### 2.2. HSF1 Down-Regulation Using shRNA

The shRNA target sequences for human HSF1 (NM_005526.4) were selected, as described in Refs [7,16]. MCF10A and MCF12A cells were transduced with lentiviruses and selected using a medium supplemented with 1 μg/mL puromycin (Life Technologies/Thermo Fisher Scientific, Waltham, MA, USA), as described in Ref [17]. Control sh.CTRL cell lines were generated by subsequent transductions of MCF10A cells with pLVX-shRNA1 vectors encoding the shRNA-SCR sequence (5′CCT AAG GTT AAG TCG CCC TCG-3′) or MCF12A cells with pLVX-shRNA1 vectors encoding the shRNA-LUC sequence (5′GTG CGT TGC TAG TAC CAAC-3′) [18].

### 2.3. HSF1 Overexpression

Human HSF1 coding sequence was amplified by PCR on the template of the hHSF1-pLNCX2 vector [16]. The sequence recognized by the EcoRI restriction enzyme was introduced into the primers. The HSF1 cDNA fragment was inserted into the pLVX-Puro plasmid (Clontech/Takara Bio, Mountain View, CA, USA) using the In-Fusion^®^ HD Cloning Kit (Clontech/Takara Bio). Infectious lentiviruses were generated by transfection of DNA into HEK293T cells, and virus-containing supernatants were collected. MCF10A or MDA-MB-468 cells were transduced with lentiviruses containing HSF1 cDNA (HSF1-pLVX) or the empty vector (pLVX) and selected using a medium supplemented with 1 μg/mL puromycin (Life Technologies).

### 2.4. Cell Proliferation

Cells (MCF10A—1.5 × 10^4^ per well; MCF12A—2 × 10^4^ per well) were seeded and cultured in 12-well plates. At the indicated time, cells were washed with PBS, fixed in cold methanol and rinsed with distilled water. Cells were stained with 0.2% crystal violet for 30 min, rinsed extensively with distilled water and dried. The cell-associated dye was extracted with 2 mL of 10% acetic acid. Aliquots (200 μL) were transferred to a 96-well plate, and the absorbance was measured at 595 nm (Synergy2 microtiter plate reader, BioTek Instruments, Winooski, VT, USA). The growth curves are shown as the absorbance ratio on days 1, 2 and 3 versus day 0 and were calculated from three to six independent experiments, each in two to three technical replicates.

### 2.5. Ki-67 Staining

MCF10A cells (5 × 10^4^) were seeded in poly-L-lysine (Sigma-Aldrich)-coated 4-well glass slides Nunc Lab-Tek II chambered coverglass (Nalge Nunc International, Rochester, NY, USA). Two days later, cells were washed with phosphate-buffered solution (PBS) and fixed with 4% paraformaldehyde for 15 min at room temperature. The cells were then washed with PBS, permeabilized with 0.1% Triton X-100 for 5 min at room temperature, washed and blocked with 2.5% normal goat serum (Abcam, Cambridge, UK). Cells were incubated with mouse monoclonal antibody against Ki-67 (1:100, MAB4190, Millipore, Burlington, MA, USA) at 4 °C overnight. The primary antibody was detected using an anti-mouse secondary antibody conjugated with AlexaFluor594 (1:200, ab150116, Abcam). Cells were counterstained using DAPI (Sigma-Aldrich) and mounted. Images were taken using a Carl Zeiss (Jena, Germany) LSM 710 confocal microscope with ZEN navigation software (v. 7.0.3.286). The experiments were performed in triplicate. Cells were counted in 10 randomly selected fields for each replicate. The Ki-67 index was represented as a ratio of Ki-67 positive cells to the overall number.

### 2.6. Senescence-Associated β-Galactosidase (SA-β-Gal) Assay

MCF10A cells (5 × 10^4^) were seeded in poly-L-lysine-coated 4-well slides (Nalge Nunc International). Two days later, cells were washed and fixed with paraformaldehyde and glutaraldehyde. Cells were washed twice with PBS and incubated at 37 °C for 16 h with a staining solution containing 40 nM citric acid/sodium phosphate (pH 6.0), 150 mM NaCl, 2 mM MgCl_2_, 1 mg/mL X-gal, 5 mM potassium ferrocyanide and 5 mM potassium ferricyanide. Cells were observed under a ZEISS AXIOPHOT microscope with ZEN navigation software.

### 2.7. Protein Extraction and Western Blotting

Whole-cell extracts and Western blot analysis were performed, as described in Ref [17]. Primary antibodies against HSF1 (1:4000, rabbit polyclonal, ADI-SPA-901, Enzo Life Sciences, Farmingdale, NY, USA), HSPA1 (1:10,000, mouse monoclonal, ADI-SPA-810, Enzo Life Sciences), ACTB (1:25,000, #A3854, Merck KGaA, Darmstadt, Germany), HSPA8/HSC70 (1:5000, #sc-7298, Santa Cruz Biotechnology, Dallas, TX, USA) and antibodies from the EMT Antibody Sampler Kit (1:1000, #9782, Cell Signaling Technology, Danvers, MA, USA) were used. The primary antibody was detected by an appropriate secondary antibody conjugated with horseradish peroxidase (Thermo Fisher Scientific) and visualized with an ECL kit (Thermo Fisher Scientific).

### 2.8. Three-Dimensional Overlay Culture and Spheroid Staining

Cells were cultured in Matrigel, as previously reported [19]. Eight-well chamber slides (Nalge Nunc International) cells were covered with growth-factor-reduced Matrigel (354230, Corning, NY, USA) and allowed to solidify in a CO_2_ incubator for 15 min. Meanwhile, cells were trypsinized, counted and diluted in DMEM/F12 culture medium containing 2% horse serum (BioWest, Nuaillé, France), 0.5 mg/mL hydrocortisone (Sigma-Aldrich), 10 µg/mL insulin, 5 ng/mL EGF (Sigma-Aldrich) and 2% Matrigel. Cells (5 × 10^3^) were overlaid on top of the solidified Matrigel and allowed to grow for 14 days (unless otherwise indicated). Spheroids were observed under the Axiovert 40 CFL microscope (Carl Zeiss), and pictures were taken. The spheroid perimeter was measured using ImageJ software (v. 1.51k). The radius of a sphere was calculated as r = C/2π (C-perimeter), and the spheroid volume was calculated as V = 3/4×π×r^3^. Spheroids formed by MCF12A cell variants were counted in 10 random fields, and their morphology was distinguished into round (oval, regular shape), disruptive (oval, irregular shape, bulgy shape) and multi-acinar (berry-shaped, multiple lobules), as described in Ref [20]. Cells were fixed with 4% paraformaldehyde, permeabilized with 0.5% Triton X-100 for 10 min at room temperature, washed with PBS containing 100 mM glycine and blocked using 10% normal goat serum (Abcam) in a buffer containing 0.1% bovine serum albumin (BSA, Sigma-Aldrich), 0.2% Triton X-100 and 0.05% Tween-20 for 1 h at room temperature. Cells were incubated with rat monoclonal antibody against ITGA6 (1:50, MAB13501, R&D Systems, MN, USA) or rabbit monoclonal antibody against CDH1 (1:100, #3195, Cell Signaling Technology) at 4 °C overnight. The primary antibody was detected using an anti-rat secondary antibody conjugated with AlexaFluor488 (1:200, Invitrogen, Waltham, MA, USA) or an anti-rabbit secondary antibody conjugated with AlexaFluor 488 (1:200, ab150077, Abcam). Cells were counterstained with TRITC-labeled phalloidin (Sigma-Aldrich) to visualize F-actin and DAPI (Sigma-Aldrich), and mounted. Images were taken using a Carl Zeiss LSM 710 confocal microscope with ZEN navigation software.

### 2.9. Global Gene Expression Profiling

Total RNA was isolated using the Direct-zol^TM^ RNA MiniPrep Kit (Zymo Research, Irvine, CA, USA) and digested with DNase I (Worthington Biochemical Corporation, Lakewood, NJ, USA). The cDNA libraries were sequenced by Illumina HighSeq 1500 (run type: paired-end; read length: 2 × 76 bp). Raw RNA-seq reads were aligned with human genome hg38 using hisat2 v 2.0.5 [21]. The aligned files were processed using Samtools (v. 1.13) [22]. Furthermore, reads aligned in the coding regions of the genome were counted using FeatureCounts (v. 1.6.5) [23] with a relevant Ensembl genes transcriptome reference, suitable for hg38. Finally, read counts were normalized using DESeq2 (v. 1.32.0) [24]. The normalized expression values were subject to differential analysis (mean-based fold change) in the R/Bioconductor programming environment. Changes (MCF10A HSF1-pLVX versus MCF10A wild type) were considered significant if the signal ratios were >2.0 or <0.50 in both biological replicates. Raw RNA-seq data were deposited in the NCBI GEO database, acc. no. GSE241795. The enrichment analyses of gene sets were performed in PANTHER Pathways using the PANTHER Overrepresentation Test, version 17.0 [25].

### 2.10. Boyden Chamber Assay

Transwell chambers (with 8 µm pore size membrane, Becton Dickinson, Franklin Lakes, NJ, USA) were coated with fibronectin (10 µg/mL, Becton Dickinson). Cells were harvested by trypsinization and suspended in a serum-free Hepes-buffered medium containing 0.1% BSA, seeded at the top of the chambers and placed in wells containing culture medium supplemented with 10% fetal bovine serum. After incubation (MCF10A cells for 4 h; breast cancer cells for 6 h) at 37 °C, cells were washed with PBS, fixed with methanol and stained with 0.2% crystal violet solution. The cells on the upper surface of the inserts were removed with cotton swabs. The migrated cells were counted under a microscope in five random fields. All experiments were performed at least three times.

### 2.11. Flow Cytometry

Cells were harvested by trypsinization and counted, and 5 × 10^5^ cells were used for staining. Cells were washed twice with PBS, suspended in the Flow Cytometry Buffer and incubated with APC-conjugated antibody against EPCAM (BioLegend, San Diego, CA, USA, 5 µL/1 × 10^6^ cells) and PE-conjugated antibody against ITGA6 (BioLegend, 5 µL/1 × 10^6^ cells) or PE-conjugated antibody against CD24 (clone ML5, 311106, BioLegend, 5 µL/1 × 10^6^ cells) and FITC-conjugated antibody against CD44 (clone BJI18, 338804, BioLegend, 5 µL/1 × 10^6^ cells) for 30 min at room temperature in the dark. Cells were suspended in PBS and analyzed with flow cytometry (BD FACS CantoTM). Isotype controls were performed.

### 2.12. Cell Viability Assay

BT-549 cells (2 × 10^3^/well) were seeded in 96-well plates. In the following days, the medium was replaced with one containing DTHIB at a concentration ranging from 0 to 200 μM in DMSO (max. DMSO concentration <0.5%), and cells were incubated for 72 h. Cell viability was determined using CellTiter 96 Aqueous One Solution reagents according to the manufacturer’s protocol (Promega, Madison, WI, USA). The absorbance of the formazan product was measured (λ = 490 nm) using a microplate reader. The experiment was carried out at least three times with three replications for each concentration of the compound tested. IC50 values were determined using the Quest Graph IC50 Calculator (AAT Bioquest, Inc, https://www.aatbio.com/tools/ic50-calculator, accessed on 28 September 2021) with the option “Set minimum response to zero”.

### 2.13. Statistical Analyses

For each dataset, the normality of the distribution was assessed using the Shapiro–Wilk test. Depending on data distribution, the homogeneity of variances was verified by the Levene test or the Brown–Forsythe test. The outliers were determined using the Tuckey criterion and the QQ plot. For the analysis of differences between the compared groups with normal distribution, the quality of the mean values was verified by the ANOVA test with a pairwise comparison performed with the HSD Tukey test or the Games–Howell test depending on the homogeneity of variance. In the case of non-Gaussian distribution, the Kruskal–Wallis ANOVA was applied to verify the hypothesis on the equality of the medians with the Conover–Iman test or the Dunn test for pairwise comparisons. *p* = 0.05 was selected as a statistical significance threshold.

## 3. Results

### 3.1. Down-Regulation of HSF1 Results in Inhibition of MCF10A Cell Proliferation and Alters the Morphology of MCF12A Cells in 2D and 3D Cultures

Several studies have shown higher expression of HSF1 in more malignant cells, which supports their growth [26,27]. We found that non-tumorigenic cell lines—MCF10A and MCF12A—derived from breast tissue of patients with fibrocystic changes [28,29], both estrogen receptor-negative [30,31], also differ in the level of HSF1, which was higher (about two-fold) in the MCF12A line (Figure 1a). Despite their similar origin, these two cell lines differ in the expression of EMT markers (Figure 1b). MCF12A cells show a significantly lower expression of epithelial CDH1 (E-cadherin) and its binding partner CTNNB1 (β-catenin) than MCF10A cells, which correlates with a higher expression of SNAI2, a known CDH1 transcriptional repressor and regulator of the epithelial-to-mesenchymal transition. Moreover, MCF12A cells express mesenchymal vimentin (VIM), which is almost undetectable in MCF10A cells. These differences in the expression levels of EMT markers are reflected in cell morphology. In subconfluent and confluent 2D culture, MCF10A cells grow in clusters and exhibit mostly a cobblestone epithelial morphology, while MCF12A cells grow as a mixed population consisting of dispersed, spindle-shaped or round cells, which surround small cell clusters (Figure 1c). In Matrigel, MCF10A cells form round spheroids with a well-polarized outer layer of cells (stained for ITGA6/α6-integrin/CD49f, a biomarker of basolateral polarity, which is localized to the basal cell membrane), while MCF12A cells, in addition to round spheroids, form disruptive acini characterized with an oval irregular and protruding shape or even multi-acinar spheroids with multiple lobules with incomplete basal polarity (Figure 1d). The above characterization of the two breast epithelial cell lines shows that, unlike MCF10A cells, MCF12A cells are more heterogeneous with some features of mesenchymal cells.

To study the effect of HSF1 on the phenotype of non-tumorigenic mammary epithelial cell lines, we first down-regulated HSF1 expression in MCF10A and MCF12A cells (Figure 2a). HSF1 silencing using specific shRNA resulted in decreased proliferation of MCF10A but not MCF12A cells (Figure 2b). The slowdown in the growth of MCF10A cells after HSF1 silencing correlated well with a reduction in the number of cells expressing Ki-67 (a biomarker of proliferation) (Figure 2c). In addition, in this cell line variant, we noted a greater number of enlarged, flattened and vacuolated cells exhibiting the senescence-associated β-galactosidase (β-gal) activity (Figure 2d). More importantly, HSF1-deficient MCF10A cells partially lost their ability to form spheroids in Matrigel (and their volume was reduced; Figure 2e,f), probably due to a lower rate of proliferation. Nevertheless, the basal polarization (as confirmed by the ITGA6 staining) was the same regardless of the HSF1 level (Figure 2e), suggesting that acini polarization was not changed. In the case of MCF12A cells with reduced HSF1 expression, they began to form clusters in subconfluent 2D culture, which was not observed in cells with normal HSF1 levels (Figure 2g, upper panel). In Matrigel, a reduction in the number of disruptive or multi-acinar spheroids in favor of round spheroids was observed after HSF1 silencing (Figure 2g,h). Moreover, HSF1-deficient spheroids exhibited a better polarized outer layer of cells (stained for ITGA6) compared to spheroids with normal HSF1 levels, which were characterized by incomplete or lost basal polarity (Figure 2g). These results suggest that in cells with certain mesenchymal features (such as MCF12A), down-regulation of HSF1 can result in adopting epithelial morphology (observed in MCF10A cells). We hypothesized that HSF1 overexpression might have the opposite effect. Therefore, in the next step, we aimed to study the effect of HSF1 overexpression on the MCF10A cell phenotype.

### 3.2. HSF1 Overexpression May Promote the Epithelial-to-Mesenchymal Transition of MCF10A Cells in a Mechanism Involving CDH1 Down-Regulation

Stable HSF1 overexpression (Figure 3a) did not affect MCF10A cell proliferation (Figure 3b). However, we observed that MCF10A cells overexpressing HSF1 changed their appearance in 2D culture, partially acquiring a fibroblast-like scattered morphology (typical of MCF12A) (Figure 3c). Additionally, spheroids in Matrigel were enlarged compared to the control ones (Figure 3c,d). Nevertheless, there was no change in the basal polarity, as evidenced by ITGA6 staining (Figure 3c). Western blot analyses of EMT markers revealed that HSF1 overexpression in MCF10A cells correlated with decreased expression of CDH1 and increased expression of VIM (Figure 3a). RNA-seq analyses (Appendix A) revealed that the cadherin signaling pathway (P00012) was also the most enriched in the set of genes down-regulated in these cells (Table 1). In addition to a decrease in CDH1 expression, RNA-seq analysis also showed a reduced expression of CDH2 and 19 genes encoding protocadherins. Among the genes most severely affected in cells with HSF1 overexpression were many genes encoding cell adhesion or cytoskeleton-organizing proteins (e.g., *GJA5*, *NID2*, *TUBA1A*, *MFAP5*, *TAGLN* were inhibited; *TTC25*, *CADM3*, *IQGAP2*, *SDK2*, *CEACAM6*, *FAT2* were up-regulated; Appendix A). On the other hand, the down-regulation of HSF1 in MCF12A cells was accompanied by increased CDH1 expression and a slight reduction in the SNAI2 level, while the VIM and CTNNB1 levels were not changed (Figure 3a). We also found a lower expression of CDH1, a cell–cell contact marker, in spheroids formed by MCF10A cells overexpressing HSF1. Moreover, even after 21 days of culture, they lacked a hollow lumen (such incomplete luminal development was considered a hallmark of a premalignant state [32]). In contrast, the control spheroids (wild type or with an empty vector) formed mature acini with a hollow lumen and well-distinguished CDH1 localization at cell–cell contacts (Figure 3e). In turn, MCF12A cells with reduced expression of HSF1 were characterized by a stronger expression of CDH1 in cell–cell contacts within the formed acini compared to control cells (Figure 3f).

To determine whether the observed changes in EMT marker levels (resulting from HSF1 overexpression or down-regulation) translate into the ability of cells to migrate, we performed the Boyden chamber assay. The number of migrating cells increased in the MCF10A cell population overexpressing HSF1 (Figure 3g) but did not change after HSF1 silencing in MCF12A cells. Additionally, the time required to detach HSF1-overexpressing MCF10A cells with trypsin was shorter than that required to detach the control cells. On the contrary, MCF12A cells with a reduced level of HSF1 needed more time to be detached than control cells (Figure 3h). These results indicate that the HSF1 level affects the adhesion of cells to the surface they grow on (the higher the level of HSF1, the weaker the adhesion of cells). The above-mentioned results indicate that HSF1 may affect some features associated with the plasticity of human mammary epithelial cells; in particular, its levels negatively correlate with CDH1 levels, which influence cellular interactions.

The acquisition of a mesenchymal-like phenotype via EMT in mammary epithelial cells and breast cancer cells has been proposed to generate stem cell features [33]. To characterize the changes in mesenchymal phenotype and stem cell characteristics associated with HSF1 levels, we first assessed the differentiation status of MCF10A and MCF12A cell variants with flow cytometry analysis using EPCAM and ITGA6, as these two markers can identify the four distinct cell populations of the human normal mammary gland [34,35] (Figure 4a). These analyses confirmed that the control (WT and those expressing pLVX or shCTRL) MCF10A and MCF12A cells had different phenotypes: MCF10A were enriched in the luminal progenitor population positive for EPCAM and ITGA6 (EPCAM^pos^/ITGA6^pos^), while MCF12A cells exhibited the basal phenotype (EPCAM^low^/ITGA6^pos^; Figure 4a). HSF1 overexpression in MCF10A cells was associated with a change in phenotype from luminal progenitors to basal-like (Figure 4a). On the other hand, the down-regulation of HSF1 in MCF12A cells (which were initially basal-like) led to the enrichment of the luminal progenitors’ subpopulation. Further analysis was performed to assess the CD44^pos^/CD24^neg^ profile, which is associated with the population of stem cell-like or cancer stem cell features [36]. Unlike MCF10A cells, which are mainly CD44^pos^/CD24^pos^, MCF12A cells exhibit the CD44^pos^/CD24^neg^ phenotype (Figure 4b), which is associated with a mesenchymal phenotype. Increasing HSF1 levels in MCF10A cells led to the acquisition of the CD44^pos^/CD24^neg^ subpopulation, while decreasing HSF1 levels in MCF12A cells led to the enrichment of the CD44^low^/CD24^pos^ subpopulation (which is frequently associated with a terminally differentiated luminal phenotype [37]) and decline in the CD44^pos^/CD24^low^ subpopulation. This indicates that HSF1 level may positively correlate with the acquisition of a mesenchymal phenotype and stem cell-like characteristics in human mammary epithelial cells. Thus, manipulating the levels of HSF1 might lead to the creation of some changes in the phenotype.

### 3.3. Higher HSF1 Levels in Triple-Negative Breast Cancer Cells Correlate with a More Invasive Phenotype

Taking into account our observation that HSF1 may play a role in the EMT of human mammary epithelial cells, we examined the correlation between HSF1 levels and the phenotype of breast cancer cells. We used several breast cancer cell lines (MDA-MB-468, BT-549, MDA-MB-231, CAL120 and HCC1395) originating from patients with triple-negative breast cancer (TNBC; negative for estrogen and progesterone receptors and HER2). Additionally, all these cell lines are characterized by wild-type BRCA1 [38]. Among them, MDA-MB-468 cells showed the lowest HSF1 level, while HCC1395 cells had the highest HSF1 level (Figure 5a). Only MDA-MB-468 cells expressed CDH1 (and lacked VIM and SNAI2) (Figure 5b) and showed the CD44^pos^/CD24^pos^ (epithelial-like) phenotype (Figure 5c). The other four cell lines lacked CDH1 expression but expressed VIM and SNAI2 (Figure 5b) and were also characterized with a predominant CD44^pos^/CD24^neg^ (mesenchymal) phenotype (Figure 5c). CTNNB1 and SNAI1 had a distinct expression in all cell lines tested (Figure 5b). The epithelial-like phenotype of MDA-MB-468 cells correlated well with the lowest ability to migrate in the transwell assay, while the highest ability was detected for BT549 cells (with the highest expression of VIM and SNAI2) (Figure 5b,d). Analyzing cell growth in Matrigel, we found that two cell lines with the highest HSF1 levels (BT-549 and HCC1395), although they differed greatly in their ability to migrate, both exhibited a stellate (invasive) phenotype (Figure 5f). MDA-MB-468 cells, possessing the lowest HSF1 level and being positive for CDH1 expression, showed the mass phenotype (less invasive). In general, the ability to migrate in the Boyden chamber correlated well with the levels of mesenchymal markers, while the transition from mass phenotype to the stellate phenotype correlated well with increasing HSF1 levels.

### 3.4. HSF1 May Affect the Invasive Phenotype of Triple-Negative Breast Cancer Cells

To test whether HSF1 can change the phenotype of breast cancer cells from epithelial-like toward mesenchymal, we constructed MDA-MB-468 cells with HSF1 overexpression (Figure 6a). The expression of CDH1, CTNBB1 or SNAI1 and the epithelial-like phenotype (CD44^pos^/CD24^pos^) did not change in these cells (Figure 6a,b). However, the number of migrating cells (Boyden assay) increased (Figure 6c), and these cells also formed more dispersed spheroids in Matrigel (Figure 6d), indicating the progression of cancer cell phenotype toward a more invasive one. It is worth noting that although we did not observe changes in CDH1 expression in cells grown in 2D conditions, weaker CDH1 expression was observed in cell–cell contacts in spheroids formed by MDA-MB-468 cells overexpressing HSF1 (Figure 6e).

Next, to study the effect of HSF1 inhibition on the viability and phenotype of the most mobile and invasive TNBC cells (i.e., BT-549; see Figure 5d,e), we used the Direct Targeted HSF1 InhiBitor (DTHIB [39]). The treatment of BT-549 cells for 48 h with DTHIB at a concentration below IC50 (which was set at 7.7; Figure 7a) led to a decrease in HSF1 levels (although these changes were not statistically significant; Figure 7b) with accompanying inhibition of heat-shock-induced HSPA1 expression by approximately 20% (Figure 7c). Growth inhibition through DTHIB was more effective in Matrigel than in 2D culture. At a concentration of 5 μM, no spheroids grew, while at a concentration of 2.5 μM (which did not affect growth in 2D culture), round, tight spheroids with a reduced elongated invasive body instead of stellate spheroids formed (Figure 7d). This suggests that DTHIB may be further tested in the treatment of TNBC.

## 4. Discussion

By studying the effect of HSF1 on the phenotype of non-tumorigenic mammary epithelial cells, we found that this factor could contribute to cell plasticity. By manipulating HSF1 levels in MCF10A and MCF12A cells used in our study, we showed that increased HSF1 levels correlated with the acquisition of mesenchymal cell characteristics, while lowering HSF1 levels could reverse this process. HSF1 overexpression or inhibition also affects the phenotype of triple-negative cancer cells, mainly their invasiveness in 3D culture. HSF1 is a well-known regulator of the cell response induced by proteotoxic stress (i.e., heat shock, exposure to heavy metals and others). Studies carried out on HSF1-deficient cells and mice revealed its role in different developmental processes associated with growth [40,41], fertility [13,42], immune response and inflammation [43,44]. Lack of HSF1 also makes mice partially resistant to neoplastic transformation, for example, driven by mutated HRAS (V12D) [45] or ERBB2 overexpression [10]. Delay in ERBB2/NEU (HER2)-induced breast tumor growth in *Hsf1*-/- mice may result from inhibited cell proliferation of mammary epithelial cells [10]. We also observed that a decrease in HSF1 levels inhibited the proliferation of MCF10A cells. However, the influence of HSF1 on cell plasticity highlights its role in breast morphogenesis and cancer development. Higher endogenous HSF1 levels observed in MCF12A cells (compared to MCF10A cells) correlated with an increased number of multi-acinar structures with an incomplete basement membrane formed in Matrigel. Such multi-acinar structures recapitulate some aspects of breast tissue hyperplasia [32,46]. Therefore, elevated HSF1 levels are likely relevant to the hyperplastic changes in the breast.

We showed that the mesenchymal phenotype resulting from HSF1 overexpression in human mammary epithelial cells is associated with enhanced migratory capacity, expression of vimentin and down-regulation of CDH1, and enrichment of the population of invasive mesenchymal CD44^high^/CD24^neg^ cells. Down-regulation of CDH1 is assumed to be the main signal initiating EMT. Its expression is repressed by several transcription factors, such as TWIST, SNAI1, SNAI2, E12/E47 and SIP1. Another mechanism can involve epigenetic mechanisms [47]. We assume that HSF1 can be directly involved in the *CDH1* gene regulation, since its promoter contains the HSF1 consensus binding site. However, HSF1 bound in the promoter has been shown to have an activating rather than a repressive function [44]. Therefore, we assume that HSF1-dependent inhibition may occur at the chromatin level, especially since it was demonstrated that HSF1 can participate in chromatin organization [17,44]. Importantly, HSF1 was shown to interact with MTA1 [48], a component of the NuRD complex that, in cooperation with SNAI1, represses *CDH1* expression [47]. It was also shown that HSF1 can induce EMT by binding to the *SNAI2* promoter (a known repressor of *CDH1* expression) and activating its transcription [9]. Nevertheless, the mechanism of the inhibition of CDH1 expression by HSF1 requires further studies.

EMT results in an increase in the phenotypic and functional cell diversity in the breast. It was shown that the acquisition of mesenchymal phenotypes via EMT is accompanied by stem cell-like characteristics with the CD44^pos^/CD24^neg^ or EPCAM^neg^/ITGA6^high^ phenotype [49,50]. The CD44^pos^/CD24^neg^ mesenchymal-like phenotype is also characteristic of cancer stem cells and is associated with invasive features of breast cancer malignancy. Similar gene expression profiles of CD44^pos^/CD24^neg^ cells from normal breast and breast tumors [51] indicate that branching in breast morphogenesis and invasion events in breast cancer progression may involve similar signaling pathways. Our data indicate that HSF1 may be involved in the molecular pathways, which link EMT to the acquisition of stem cell properties. The supportive role of HSF1 for the cancer stem cell (or stem cell-like) phenotype was already demonstrated in breast cancer [52], melanoma [16], gynecological cancer [53] and acute myeloid leukemia [54]. Additionally, several experimental models have demonstrated that a high level of HSF1 was required for EMT associated with the invasion and migration of human pancreatic cancer cells [55] or hepatocellular carcinoma cells [56]. Surprisingly, in our model, the increase in HSF1 expression did not affect the stem cell-like characteristics of MD-MB-468 breast cancer cells (which exhibit CD44^pos^/CD24^pos^) but facilitated their migration and growth in Matrigel. This shows that high levels of HSF1 can regulate several steps of the invasion cascade in cancer cells, which is not necessarily linked to the acquisition of the CD44^pos^/CD24^neg^ phenotype. Many experimental data indicate that HSF1 is indispensable for cancer cell migration and invasion, which plays an important role in cancer progression [55,57,58,59]. Importantly, we observed that inhibition of HSF1 using its specific inhibitor, DTHIB [39], reduced the proliferation of highly invasive BT-549 cells and their ability to invade Matrigel.

The important issue, which remains unresolved, is the regulation of HSF1 levels in cells. Analyses of HSF1 levels in murine tissues revealed a high expression in the testes and a weaker expression in the brain, ovary, heart, lung and spleen [60]. According to the Human Protein Atlas portal (proteinatlas.org), there is also variation in HSF1 levels in human tissues, with high expression in some regions of the brain, lung, testes and breast (and others) and lower expression in the liver and soft tissues. Early reports showed that human HSF1 is constitutively transcribed under non-shock conditions, and the level of mRNA did not change in heat-shocked HeLa cells compared to control ones [61]. Later data demonstrated increased levels of HSF1 mRNA in many cancer types [27,62], suggesting regulation of HSF1 expression at the transcriptional level. NOTCH binding to the HSF1 promoter was demonstrated in samples from patients with acute T-cell lymphoblastic leukemia, resulting in increased expression of HSF1 and its downstream effectors with subsequent activation of the cellular stress response machinery [63]. Additionally, five transcription factors, such as MYC, MXI1, MAX1, TAF1 and YY1, were found to bind around the transcription start site of the *HSF1* gene [64]. In particular, the MYC oncogene could be responsible for higher levels of HSF1 expression in tumors (according to the Human Protein Atlas, the expression levels of both genes in tumors correlate well with each other). HSF1 can also be up-regulated at the transcriptional level by NFE2L2 during oxidative stress, which is relatively higher in cancer cells [65]. However, HSF1 activation appears to be primarily regulated at the protein level [64,66]. HSF1 is a long-lived protein with a half-life of approximately 14–20 h, which allows it to exert an immediate cytoprotective effect [67]. The regulation of its activity is complex, can be cell-type specific and is not yet fully understood. It is possible that changes in HSF1 levels in mammary epithelial cells can promote gland differentiation during pregnancy and lactation. This may be one of the signaling mechanisms, which regulate the continued plasticity of the mammary epithelium, but cancer cells may exploit or corrupt it to support aberrant growth and progression toward invasion.

## 5. Conclusions

Heat shock factor 1 is a well-known regulator of cell response to environmental stress. We found that an increase in its expression may affect the intrinsic ability of human mammary epithelial cells to acquire mesenchymal characteristics. Elevated levels of HSF1 also positively correlate with invasive characteristics of triple-negative breast cancer cells. The use of DTHIB, a low-molecular-weight HSF1 inhibitor, significantly inhibits the growth of the highly invasive BT549 cell line in standard culture or Matrigel. We postulate that this inhibitor can be tested as an additional treatment for patients with TNBC.

## Figures and Tables

**Figure 1 cancers-15-04965-f001:**
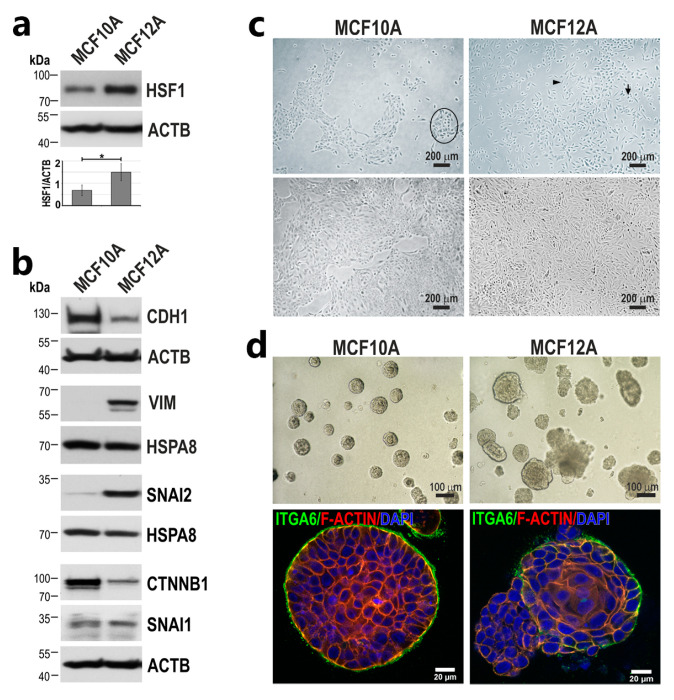
MCF10A and MCF12A cells derived from the breast tissue of patients with fibrocystic changes differ in HSF1 and EMT marker levels and morphology. (**a**) The expression of HSF1 and (**b**) EMT markers was assessed by Western blot. ACTB and HSPA8 were used as loading controls. The graph shows the results of densitometric analyses (n = 3). * *p* < 0.05 (significance of differences). (**c**) Morphology of cells in 2D culture: phase-contrast images of subconfluent (upper panel) and confluent cells (bottom panel). An example cluster of cells is marked with an ellipse; the arrow points to a round cell, and the arrowhead points to a spindle cell. Scale bar 200 µm. (**d**) Morphology of cells cultured in Matrigel: phase-contrast images (upper panel; scale bar 100 µm) and confocal images of spheroids stained for ITGA6, F-actin and DNA (bottom panel; scale bar 20 µm). Original western blots are presented in Appendix A.

**Figure 2 cancers-15-04965-f002:**
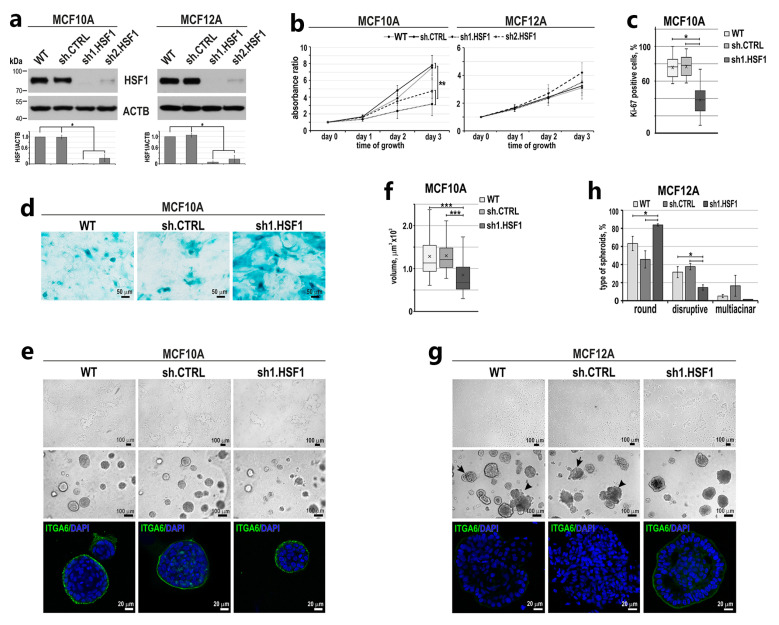
Effect of HSF1 down-regulation on the growth and ability to differentiate of MCF10A and MCF12A cells. (**a**) Western blot analysis of HSF1 expression in either unmodified (WT) cells and cells transduced with lentiviruses containing shRNAs: control (sh.CTRL) or targeting HSF1 (sh1.HSF1, sh2.HSF1). Actin (ACTB) was used as a protein loading control. The graphs below show the results of densitometric analyses (n = 3). (**b**) Proliferation rates (assessed with crystal violet staining) of MCF10A (n = 5) and MCF12A (n = 3) cell variants. (**c**) The number of Ki-67 positive cells in MCF10A cell variants (n = 3). (**d**) Senescence-associated β-galactosidase activity (yielding blue color) analyzed in MCF10A cell variants (n = 3). Scale bar 50 μm. Morphology of MCF10A (**e**) MCF12A (**g**) cell variants cultured in 2D (upper panels) and Matrigel (middle and bottom panels). Phase-contrast images (upper and middle panels; scale bar 100 µm) and confocal images of spheroids stained for ITGA6 and DNA (bottom panel; scale bar 20 µm). Arrows point to representative acini with disruptive morphology, and arrowheads indicate representative acini with multi-acinar morphology. (**f**) The volume of spheroids formed in Matrigel by MCF10A cell variants. Boxplots represent the median, upper and lower quartiles, maximum and minimum. Statistically significant differences between unmodified cells, cells with empty vector or cells overexpressing HSF1 are marked. (**h**) Morphology of spheroids formed by MCF12A cell variants was distinguished as normal, disruptive and multi-acinar, as described by Ref [20]. *** *p* < 0.001, ** *p* < 0.01, * *p* < 0.05 (significance of differences). Original western blots are presented in Appendix A.

**Figure 3 cancers-15-04965-f003:**
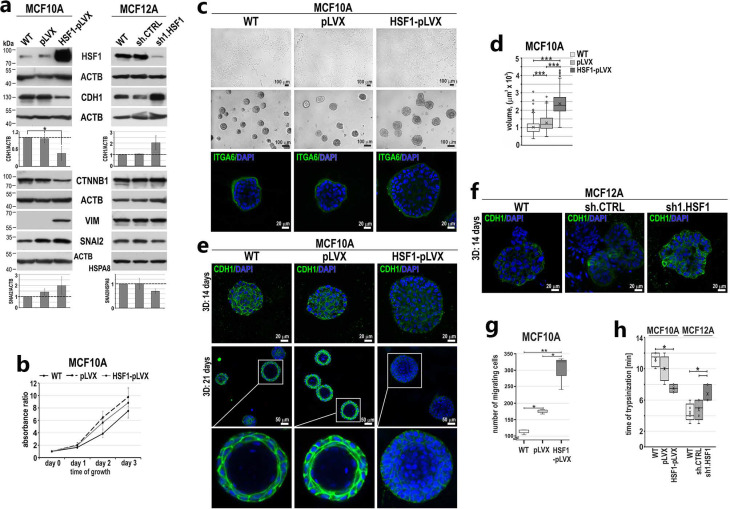
HSF1 levels affect the expression of EMT markers and phenotype of MCF10A and MCF12A cells. (**a**) Western blot analysis of HSF1 and EMT markers in cells with overexpression (MCF10A) or knockdown (MCF12A) of HSF1. WT, unmodified cells; pLVX or shCTRL, cells transduced with lentiviruses containing the empty vector or control shRNA; HSF1-pLVX or sh1.HSF1, cells transduced with lentiviruses containing the HSF1 cDNA or shRNA targeting HSF1. ACTB and HSPA8 were used as loading controls. Graphs show the results of densitometric analyses (n = 3). (**b**) Proliferation rates assessed through crystal violet staining in MCF10A cell variants (n = 5). (**c**) Morphology of MCF10A cell variants in 2D culture (subconfluent; upper panel) and Matrigel (middle and bottom panels). Phase-contrast images (upper and middle panels; scale bar 100 µm) and confocal images of spheroids stained for ITGA6 and DNA (bottom panel). (**d**) The volume of spheroids formed in Matrigel by MCF10A cell variants. Boxplots represent the median, upper and lower quartiles, maximum and minimum. (**e**,**f**) Confocal images of spheroids grown in Matrigel for 14 or 21 days and stained for CDH1 and DNA (scale bar 20 µm and 50 µm, respectively). (**g**) Migratory capacity of MCF10A cell variants analyzed with Boyden chamber assay (n = 3). (**h**) Time of cell detachment from the plastic surface by trypsinization. Boxplots represent the median, upper and lower quartiles, maximum and minimum. *** *p* < 0.001, ** *p* < 0.01, * *p* < 0.05 (significance of differences). Original western blots are presented in Appendix A.

**Figure 4 cancers-15-04965-f004:**
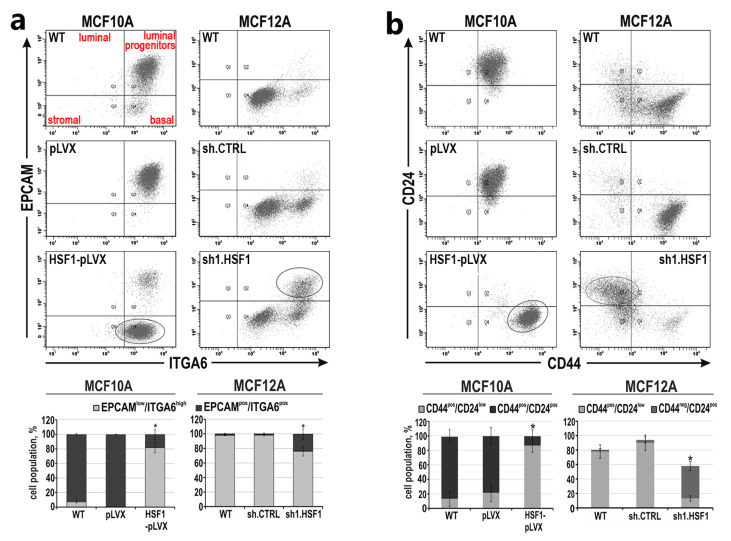
Mesenchymal and stem cell-like characteristics of MCF10A and MCF12A cell variants with altered HSF1 expression. (**a**) Subpopulations defined by expression of EPCAM and ITGA6 or (**b**) CD24 and CD44 using flow cytometry. The percentages of subpopulations shown in the bar graphs are the mean ± SEM of three independent experiments. Ovals point out cell populations, which are enriched when HSF1 expression levels are altered. * *p* < 0.05 (significance of differences).

**Figure 5 cancers-15-04965-f005:**
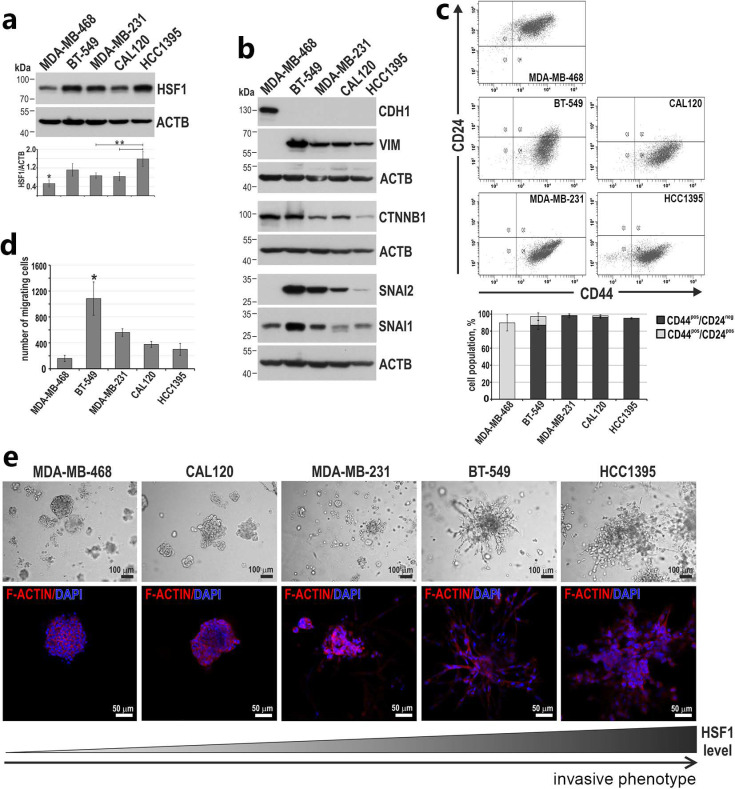
HSF1 levels in triple-negative breast cancer cell lines positively correlate with the acquisition of invasive phenotype in Matrigel. (**a**) Western blot analysis of HSF1 and (**b**) EMT markers. ACTB was used as a loading control. The graph shows the results of the densitometric analyses (n = 3). (**c**) Subpopulations defined by the expression of CD24 and CD44 using flow cytometry. The percentages of subpopulations shown on the bar graphs are the mean ± SEM of three independent experiments. (**d**) Cell migration ability analyzed using the Boyden chamber assay (n = 3). (**e**) Morphology of cells grown in Matrigel (ranked from most regular to most diffuse). Phase-contrast images (upper panel) and confocal images of spheroids stained for F-actin and DNA (bottom panel). ** *p* < 0.01, * *p* < 0.05 (significance of differences). Original western blots are presented in Appendix A.

**Figure 6 cancers-15-04965-f006:**
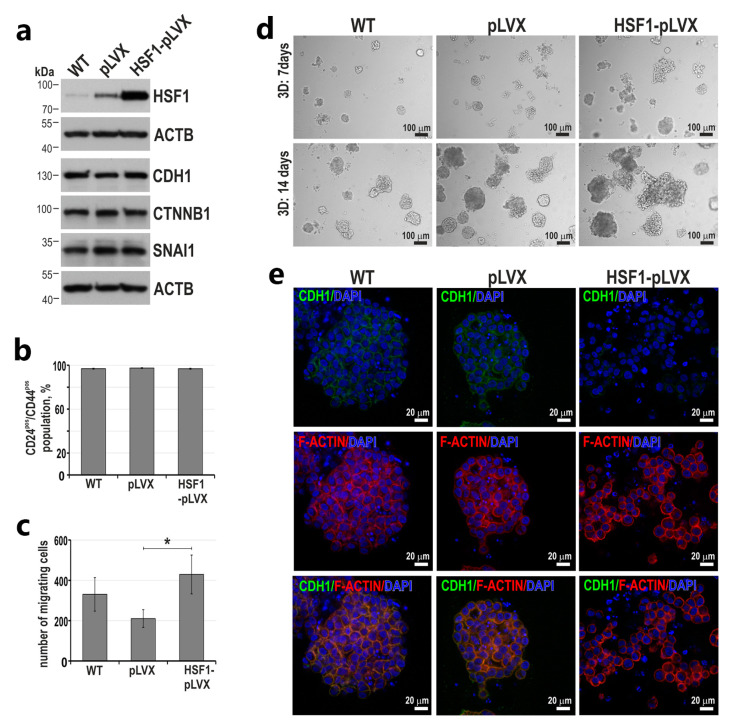
Characterization of MDA-MB-468 cells overexpressing HSF1. (**a**) Western blot analysis of HSF1 and EMT markers. WT, unmodified cells; pLVX, cells transduced with lentiviruses containing the empty vector; HSF1-pLVX, cells transduced with lentiviruses containing the HSF1 cDNA. ACTB was used as a loading control. (**b**) Percentages of the CD24+/CD44+ subpopulations in cell variants defined by flow cytometry (the mean ± SEM of three independent experiments). (**c**) The cell migration ability was analyzed with the Boyden chamber assay (n = 3). * *p* < 0.05 (significance of differences). (**d**) The morphology of cell variants grown in Matrigel for 7 or 14 days; phase-contrast images (scale bar 100 µm). (**e**) Confocal images of spheroids grown in Matrigel for 14 days and stained for CDH1, F-actin and DNA (scale bar 20 µm). Original western blots are presented in Appendix A.

**Figure 7 cancers-15-04965-f007:**
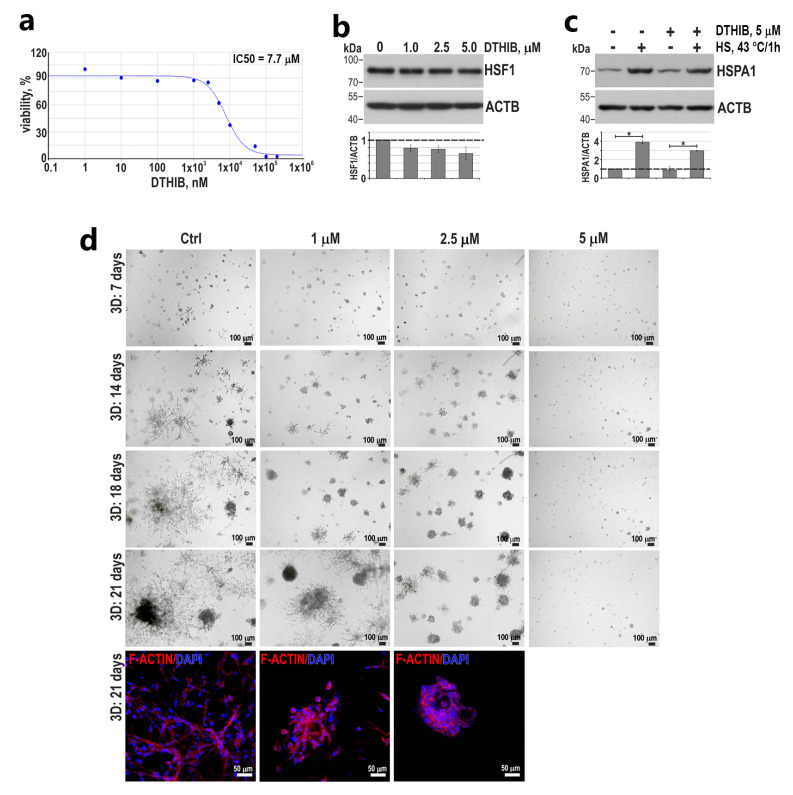
The effect of HSF1 inhibitor, DTHIB, on BT-549 cell proliferation and growth in Matrigel. (**a**) Viability of cells treated with DTHIB for 72 h and assessed with MTS. IC50 plots and values were generated with the Quest Graph IC50 Calculator. (**b**) Western blot analysis of HSF1 expression in cells treated with DTHIB for 48 h. ACTB was used as a loading control. The graphs below show the results of densitometric analyses (n = 3). (**c**) Western blot analysis of HSPA1 expression in cells pretreated with 5 μM DTHIB for 48 h before heat shock (HS, 43 °C for 1 h and recovery at 37 °C for 6 h). ACTB was used as a loading control. The graph shows the results of densitometric analyses (n = 3). * *p* < 0.05 (significance of differences). (**d**) The morphology of cells grown in Matrigel for 7–21 days in the presence of DTHIB; phase-contrast images. The bottom panels present the confocal images of spheroids grown in Matrigel for 21 days and stained for F-actin and DNA (scale bar 50 µm). Original western blots are presented in Appendix A.

**Table 1 cancers-15-04965-t001:** The enrichment of pathways in sets of genes down-regulated and up-regulated at least two-fold (RNA-seq analysis) in MCF10A cells overexpressing HSF1 in relation to wild-type cells was tested in PANTHER Pathways using the PANTHER Overrepresentation Test (PANTHER version 17.0 released 22 February 2022). Test Type: FISHER; Correction: FDR.

PANTHER Pathways	No. of Genes in the Reference Set	No. of Affected Genes		Fold Enrichment	Raw *p*-Value	FDR
		Observed	Expected			
Down-regulated (total: 1397)						
Cadherin signaling pathway (P00012)	166	33	11.14	2.96	2.98 × 10^7^	2.39 × 10^5^
Alzheimer disease–presenilin pathway (P00004)	131	24	8.79	2.73	3.81 × 10^5^	2.03 × 10^3^
Wnt signaling pathway (P00057)	314	40	21.08	1.90	3.50 × 10^4^	1.40 × 10^2^
Up-regulated (total: 1106)						
Serine glycine biosynthesis (P02776)	6	4	0.32	12.31	1.13 × 10^3^	3.63 × 10^2^
DNA replication (P00017)	29	13	1.57	8.28	1.03 × 10^7^	1.64 × 10^5^
p53 pathway feedback loops 2 (P04398)	51	10	2.76	3.62	1.02 × 10^3^	4.06 × 10^2^
p53 pathway (P00059)	89	14	4.82	2. 90	7.89 × 10^4^	4.21 × 10^2^
Up- and down-regulated (total: 2503)						
DNA replication (P00017)	29	13	3.51	3.70	3.28 × 10^4^	1.31 × 10^2^
Alzheimer disease–presenilin pathway (P00004)	131	34	15.86	2.14	2.06 × 10^4^	1.65 × 10^2^
Cadherin signaling pathway (P00012)	166	40	20.09	1.99	2.53 × 10^4^	1.35 × 10^2^

## Data Availability

The data presented in this study are openly available in the NCBI GEO database, acc. no. GSE241795.

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
