# Peer review of "An Increase in HSF1 Expression Directs Human Mammary Epithelial Cells toward a Mesenchymal Phenotype"

_cancers, 2023, doi:10.3390/cancers15204965_

Round 1
Reviewer 1 Report
It is an interesting MS with sufficient experiments to support the findings
But how this would apply to human breast development and cancer?
The concept of oncogenesis of HSF1 has been published in many reports.
What is proteotoxic stress in humans would trigger breast cancer?
The significance of the experiment to human need to be addressed. The author may include this ref
: Dai C. 2017 The heat-shock, or HSF1-mediated proteotoxic stress, response in cancer: from proteomic stability to oncogenesis. Phil. Trans. R. Soc. B 373: 20160525. http://dx.doi.org/10.1098/rstb.2016.052
minor edit
Reviewer 2 Report
This article is related to one of the most important process which impact cancerogenesis in many cancers, including breast cancer, epithelial-to-mesenchymal transition. Some EMT drivers are well known for a long time but the vast majority of them have yet to be investigated. This article scrutinizses comprehensively the well known protein HSF1 as a driver for EMT in breast cancer. The article is well organised and involves different methods to show the EMT effect of HSF1 and discuss its potential therapeutic role in target treatment.
Reviewer 3 Report
Authors present compelling data in cells and spheroids for the use of DTHIB in triple-negative breast cancer. The rationale for the application of this inhibitor in triple-negative breast cancer lines seems lacking. Authors should use existing literature or cancer repositories such as TCGA on platforms such as cbioportal and see how the expression of HSF1 varies in patients of triple-negative breast cancer with markers of EMT and disease outcome. DTHIB has been suggested for the use drug drug-resistant tumors. It would also strengthen the manuscript a lot if the authors included a cell proliferation experiment including the existing chemotherapeutic agents such as Doxorubicin in combination with DTHIB.
Reviewer 4 Report
Manuscript has a good title. English language has good quality. Some figures should be changed in the manuscript.
1. About 3, line 102-106
Why you implemented hydrocortisone and insulin in the culture medium of MCF10A?
2. About Figure 1
+ in Figure 1C and D: please determine some important details of this figure
+ Figure 1: please make the scales of figure bigger and more visible
3. Abput line 372-374 in page 9
Why you have mentioned this sentence?
"HSF1 may affect some features associated
with the plasticity of human mammary epithelial cells, especially its levels negatively correlate with CDH1 levels, which influence cellular interactions."
4. Please reorganize the place of each figures in the manuscript, some of them are placed in improper site (they are located in the middle of a paragraph)
5. About line 417 in page 11
You have mentioned that "By manipulating HSF1 levels, the phenotype of these cells can be altered"
Please change this sentence. The sentence "manipulating the levels of HSF1 might lead to create some changes in the phenotype" seems to be more proper.
6. About line 464-467 in page 13
Why you have mebtioned that " these cells acquired a grape-like phenotype in matrigel (Figure 6d), indicating the progression of cancer cell phenotype toward a more invasive one"? What is the connection between grape-like phenotype and progression of cancer cell phenotype?
7. About page 15, line 505-506
Please discuss about the probable mechanisms of involving HSF1 in cell plasticity based on prior surveys (If possible, please insert a simple figure)
8. Please check and adjust the "Reference list" based on the regulations of reference list of journal. (Titles, doi, the name of journal and ... )
Round 2
Reviewer 1 Report
It is not clear about the postulation about applying to TNBC in the conclusion
Reviewer 4 Report
I dont have any more comments.